# Learning Not to Learn: Nature versus Nurture in Silico

## Abstract

Animals are equipped with a rich innate repertoire of sensory, behavioral and motor skills, which allows them to interact with the world immediately after birth. At the same time, many behaviors are highly adaptive and can be tailored to specific environments by means of learning. In this work, we use mathematical analysis and the framework of memory-based meta-learning (or 'learning to learn') to answer when it is beneficial to learn such an adaptive strategy and when to hard-code a heuristic behavior. We find that the interplay of ecological uncertainty, task complexity and the agents' lifetime has crucial effects on the meta-learned amortized Bayesian inference performed by an agent. There exist two regimes: One in which meta-learning yields a learning algorithm that implements task-dependent information-integration and a second regime in which meta-learning imprints a heuristic or 'hard-coded' behavior. Further analysis reveals that non-adaptive behaviors are not only optimal for aspects of the environment that are stable across individuals, but also in situations where an adaptation to the environment would in fact be highly beneficial, but could not be done quickly enough to be exploited within the remaining lifetime. Hard-coded behaviors should hence not only be those that always work, but also those that are too complex to be learned within a reasonable time frame.

## 1 Introduction

The *'nature versus nurture'* debate (e.g., Mutti et al., 1996; Tabery, 2014) – the question of which aspects of behavior are 'hard-coded' by evolution, and which are learned from experience – is one of the oldest and most controversial debates in biology. Evolutionary principles prescribe that hard-coded behavioral routines should be those for which there is no benefit in adaptation. This is believed to be the case for behaviors whose evolutionary advantage varies little among individuals of a species. Mating instincts or flight reflexes are general solutions that rarely present an evolutionary disadvantage. On the other hand, features of the environment that vary substantially for individuals of a species potentially ask for adaptive behavior (Buss, 2015). Naturally, the same principles should not only apply to biological but also to artificial agents. But how can a reinforcement learning agent differentiate between these two behavioral regimes?

A promising approach to automatically learn rules of adaptation that facilitate environment-specific specialization is meta-learning (Schmidhuber, 1987; Thrun & Pratt, 1998). At its core lies the idea of using generic optimization methods to learn inductive biases for a given ensemble of tasks. In this approach, the inductive bias usually has its own set of parameters (e.g., weights in a recurrent network; Hochreiter et al., 2001) that are optimized on the whole task ensemble, that is, on a long, 'evolutionary' time scale. These parameters in turn control how a different set of parameters (e.g., activities in the network) are updated on a much faster time scale. These rapidly adapting parameters then allow the system to adapt to a specific task at hand. Notably, the parameters of the system that are subject to 'nature' – i.e., those that shape the inductive bias and are common across tasks – and those that are subject to 'nurture' are usually predefined from the start.

In this work, we use the memory-based meta-learning approach for a different goal, namely to acquire a qualitative understanding of which aspects of behavior should be hard-coded and which should be adaptive. Our hypothesis is that meta-learning can not only learn efficient learning algorithms, but can also decide not to be adaptive at all, and to instead apply a generic heuristic to the whole

ensemble of tasks. Phrased in the language of biology, meta-learning can decide whether to hard-code a behavior or to render it adaptive, based on the range of environments the individuals of a species could encounter.

We study the dependence of the meta-learned algorithm on three central features of the meta-reinforcement learning problem:

- **Ecological uncertainty**: How diverse is the range of tasks the agent could encounter?
- **Task complexity**: How long does it take to learn the optimal strategy for the task at hand? Note that this could be different from the time it takes to execute the optimal strategy.
- **Expected lifetime**: How much time can the agent spend on exploration and exploitation?

Using analytical and numerical analyses, we show that non-adaptive behaviors are optimal in two cases – when the optimal policy varies little across the tasks within the task ensemble and when the time it takes to learn the optimal policy is too long to allow a sufficient exploitation of the learned policy.

Our results suggest that not only the design of the meta-task distribution, but also the lifetime of the agent can have strong effects on the meta-learned algorithm of RNN-based agents. In particular, we find highly nonlinear and potentially discontinuous effects of ecological uncertainty, task complexity and lifetime on the optimal algorithm. As a consequence, a meta-learned adaptation strategy that was optimized, e.g., for a given lifetime may not generalize well to other lifetimes. This is essential for research questions that are interested in the conducted adaptation behavior, including curriculum design, safe exploration as well as human-in-the-loop applications. Our work may provide a principled way of examining the constraint-dependence of meta-learned inductive biases.

The remainder of this paper is structured as follows: First, we review the background in memory-based meta-reinforcement learning and contrast the related literature. Afterwards, we analyze a Gaussian multi-arm bandit setting, which allows us to analytically disentangle the behavioral impact of ecological uncertainty, task complexity and lifetime. Our derivation of the lifetime-dependent Bayes optimal exploration reveals a highly non-linear interplay of these three factors. We show numerically that memory-based meta-learning reproduces our theoretical results and can learn *not to learn*. Furthermore, we extend our analysis to more complicated exploration problems. Throughout, we analyze the resulting recurrent dynamics of the network and the representations associated with learning and non-adaptive strategies.

## 2 RELATED WORK & BACKGROUND

Meta-learning or 'learning to learn' (e.g., Schmidhuber, 1987; Thrun & Pratt, 1998; Hochreiter et al., 2001; Duan et al., 2016; Wang et al., 2016; Finn et al., 2017) has been proposed as a computational framework for acquiring task distribution-specific learning rules. During a costly outer loop optimization, an agent crafts a niche-specific adaptation strategy, which is applicable to an engineered task distribution. At inference time, the acquired inner loop learning algorithm is executed for a fixed amount of timesteps (lifetime) on a test task. This framework has successfully been applied to a range of applications such as the meta-learning of optimization updates (Andrychowicz et al., 2016; Flennerhag et al., 2018; 2019), agent (Rabinowitz et al., 2018) and world models (Nagabandi et al., 2018) and explicit models of memory (Santoro et al., 2016; Bartunov et al., 2019). Already, early work by Schmidhuber (1987) suggested an evolutionary perspective on recursively learning the rules of learning. This perspective holds the promise of explaining the emergence of mechanisms underlying both natural and artificial behaviors. Furthermore, a similarity between the hidden activations of LSTM-based meta-learners and the recurrent activity of neurons in the prefrontal cortex (Wang et al., 2018) has recently been suggested.

Previous work has shown that LSTM-based meta-learning is capable of distilling a sequential integration algorithm akin to amortized Bayesian inference (Ortega et al., 2019; Rabinowitz, 2019; Mikulik et al., 2020). Here we investigate when the integration of information might not be the optimal strategy to meta-learn. We analytically characterize a task regime in which not adapting to sensory information is optimal. Furthermore, we study whether LSTM-based meta-learning is capable of inferring when to learn and when to execute a non-adaptive program. Rabinowitz (2019)

previously studied the outer loop learning dynamics and found differences across several tasks, the origin of which is however not fully understood. Our work may provide an explanation for these different meta-learning dynamics and the dependence on the task distribution as well as the time horizon of adaptation.

Our work is most closely related to Pardo et al. (2017) and Zintgraf et al. (2019). Pardo et al. (2017) study the impact of fixed time limits and time-awareness on deep reinforcement learning agents. They propose using a timestamp as part of the state representation in order to avoid state-aliasing and the non-Markovianity resulting from a finite horizon treatment of an infinite horizon problem. Our setting differs in several aspects. First, we study the case of meta-reinforcement learning where the agent has to learn within a single lifetime. Second, we focus on a finite horizon perspective with limited adaptation. Zintgraf et al. (2019), on the other hand, investigate meta reinforcement-learning for Bayes-adaptive Markov Decision Processes and introduce a novel architecture that disentangles task-specific belief representations from policy representations. Similarly to our work, Zintgraf et al. (2019) are interested in using the meta-learning framework to distill Bayes optimal exploration behavior. While their adaptation setup extends over multiple episodes, we focus on single lifetime adaption and analytically analyze when it is beneficial to learn in the first place.

Finally, our work extends upon the efforts of computational ethology (Stephens, 1991) and experimental evolution (Dunlap & Stephens, 2009; 2016; Marcus et al., 2018), which aims to characterize the conditions under which behavioral plasticity may evolve. Their work shows that both environmental change and the predictability of the environment shape the selection pressure, which evolves adaptive traits. Our work is based on memory-based meta-learning with function approximation and aims to extend these original findings to task distributions for which no analytical solution may be available.

## 3    LEARNING NOT TO LEARN

To disentangle the influence of ecological uncertainty, task complexity, and lifetime on the nature of the meta-learned strategy, we first focus on a minimal two-arm Gaussian bandit task, which allows for an analytical solution. The agent experiences episodes consisting of $T$ arm pulls, representing the lifetime of the agent. The statistics of the bandit are constant during each episode, but vary between episodes. To keep it simple, one of the two arms is deterministic and always returns a reward of 0. The task distribution is represented by the variable expected reward of the other arm, which is sampled at the beginning of an episode, from a Gaussian distribution with mean -1 and standard deviation $\sigma_p$, i.e. $\mu \sim \mathcal{N}(-1, \sigma_p^2)$. The standard deviation $\sigma_p$ controls the uncertainty of the ecological niche. For $\sigma_p \ll 1$, the deterministic arm is almost always the better option. For $\sigma_p \gg 1$, the chances of either arm being the best in the given episode are largely even. While the mean $\mu$ remains constant for the lifetime $T$ of the agent, the reward obtained in a given trial is stochastic and is sampled from a second Gaussian, $r \sim \mathcal{N}(\mu, \sigma_l)$. This trial-to-trial variability controls how many pulls the agent needs to estimate the mean reward of the stochastic arm. The standard deviation $\sigma_l$ hence controls how quickly the agent can learn the optimal policy. We therefore use it as a proxy for task complexity.

In this simple setting, the optimal meta-learned strategy can be calculated analytically. The optimal exploration strategy is to initially explore the stochastic arm for a given trial number $n$. Afterwards, it chooses the best arm based on its maximum a posteriori-estimate of the remaining episode return. The optimal amount of exploration trials $n^\star$ can then be derived analytically: [1]

$$n^\star = \arg\max_n \mathbb{E}[\sum_{t=1}^{T} r_t | n, T, \sigma_l, \sigma_p] = \arg\max_n \left[ -n + \mathbb{E}_{\mu,r} \left[ (T - n) \times \mu \times p(\hat{\mu} > 0) \right] \right] ,$$

where $\hat{\mu}$ is the estimate of the mean reward of the stochatic arm after the $n$ exploration trials. We find two distinct types of behavior (left-hand side of figure 1): A regime in which learning via exploration is effective and a second regime in which not learning is the optimal behavior. It may be optimal not to learn for two reasons: First, the ecological uncertainty may be so small that it is very unlikely that the stochastic first arm is better. Second, if the trial-to-trial variability is too large relative to the range

---

[1] Please refer to the supplementary material for a detailed derivation of this analytical result as well as the hyperparameters of the numerical experiments.

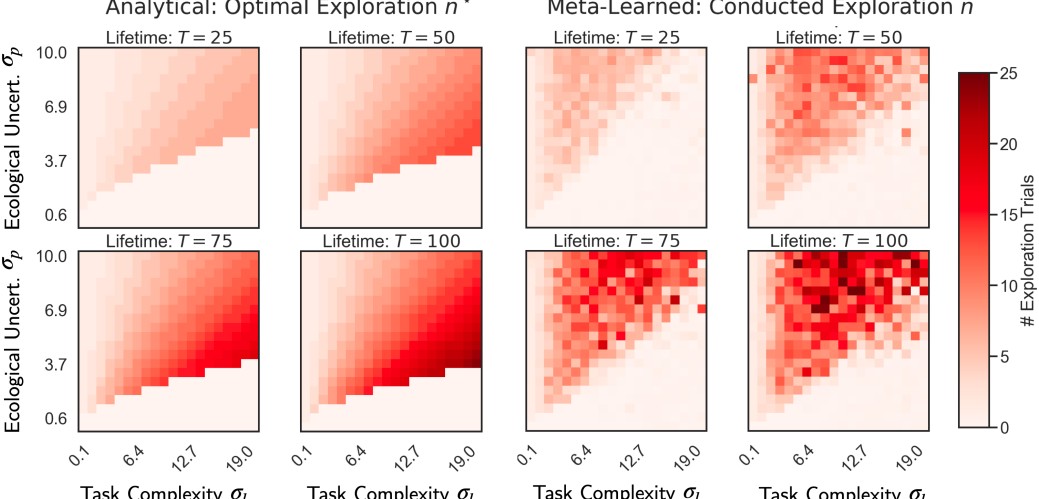

Figure 1: Theory and meta-learned exploration in a two-arm Gaussian bandit. **Left**: Bayes optimal exploration behavior for different lifetimes and across uncertainty conditions $\sigma_l, \sigma_p$. **Right**: Meta-learned exploration behavior using the RL$^2$ (Wang et al., 2016) framework. There exist two behavioral regimes (learning by exploration and a heuristic non-explorative strategy) for both the theoretical result and the numerical meta-learned behaviors. The amount of meta-learned exploration is averaged both over 5 independent training runs and 100 episodes for each of the 400 trained networks.

of potential ecological niches, so that it may simply not be possible to integrate sufficient information given a limited lifespan. We make two observations:

1. There exists a hard nonlinear threshold between learning and not learning behaviors described by the ratio of $\sigma_l$ and $\sigma_p$. If $\sigma_l$ is too large, the value of exploration (or the reduction in uncertainty) is too small to be profitable within the remaining lifetime of the agent. Instead, it is advantageous to hard-code a heuristic choice.

2. The two regimes consistently exist across different lifetimes. As the lifetime grows, the learning regime becomes more and more prevalent. Given a sufficient amount of time, learning by exploring the uncertain arm is the best strategy.

Is the common meta-learning framework capable of reproducing these different qualitative behaviors and performing Bayes optimal amortized inference across the entire spectrum of meta-task distributions? Or differently put: Can memory-based meta-learning yield agents that do not only learn to learn but that also learn *not* to learn? To answer this question, we train LSTM-based RL$^2$ (Wang et al., 2016) agents with the standard synchronous actor-critic (Mnih et al., 2016) setup on the same grid of ecological uncertainties $\sigma_p$ and "task complexities" $\sigma_l$. The input $x_t$ to the network at time $t$ consists of the action of the previous timestep, a monotonically increasing timestamp within the current episode and crucially the reward of the previous timestep, $x_t = \{a_{t-1}, \phi(t), r_{t-1}\}$. The recurrent weight dynamics of the inner loop can then implement an internal learning algorithm that integrates previous experiences. After collecting a set of trajectories, we optimize the weights and initial condition of the hidden state with an outer loop gradient descent update to minimize the common actor-critic objective.

We obtain the amount of meta-learned exploration by testing the RL$^2$ agents on hold-out bandits for which we set $\sigma_p = 0$ and only vary $\sigma_l$. Thereby, it is ensured that the deterministic arm is the better arm. We can then define the number of exploration trials as the pulls from the suboptimal stochastic arm. We observe that meta-learning is capable of yielding agents that behave according to our derived theory of a Bayes optimal agent, which explicitly knows the given lifetime as well as uncertainties $\sigma_l, \sigma_p$ (figure 1). Importantly, the meta-learned behavior also falls into two regimes: A regime in which the meta-learned strategy resembles a learning algorithm and a regime in which the recurrent dynamics encode a hard-coded choice of the deterministic arm. Furthermore, the edge between the

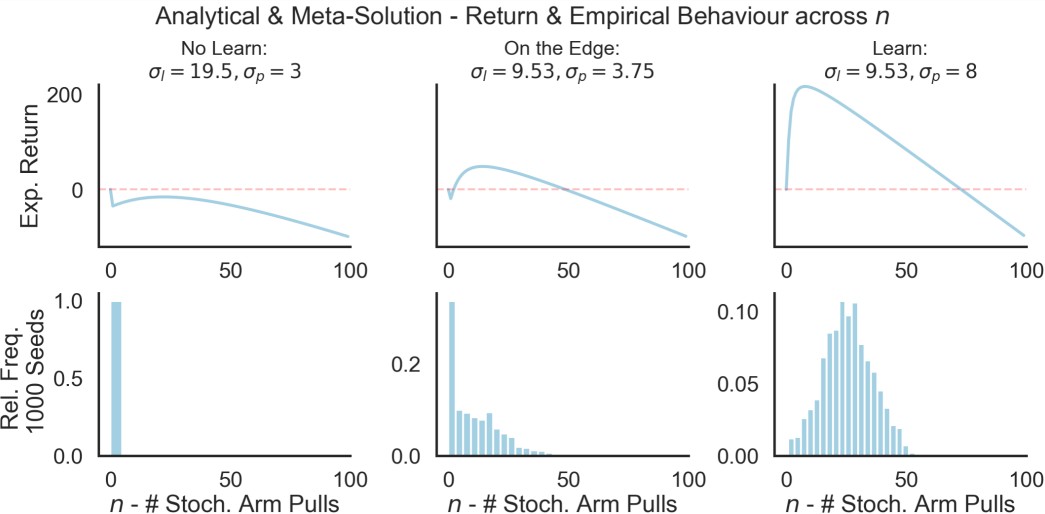

Figure 2: Bimodality of the reward landscape. **Upper row**. The Bayesian model predicts a bimodal dependence of the expected return on the policy. Parameters for lifetime $T = 100$ in the non-learning (left) and learning regime (right) and close the transition edge (middle). **Bottom row.** Distribution of the mean number of explorative pulls in 1000 separatively trained network with different random seeds. Close to the edge, the networks fall into two classes: networks either abandon all exploration (peak at $n = 0$) or explore and learn. Away from the transition, all 1000 networks adopt a similar strategy.

two meta-learned regimes shifts with the agent's lifetime as predicted by the Bayesian theory. As the lifetime increases, wider ecological niches at higher levels of task complexity become solvable and the strategy of learning profitable.

In the Bayesian model, the edge between the two regimes is located at parameter values where the learning strategy and the non-learning strategy perform equally well. Because these two strategies are very distinct, we wondered whether the reward landscape for the memory-based meta-learner has two local maxima corresponding to the two strategies (figure 2). To test this, we trained $N = 1000$ networks with different initial conditions, for task parameters close to the edge, but in the regime where the theoretically optimal strategy would be to learn. We then evaluated for each network the number of explorative pulls of the stochastic arm, averaged across 100 episodes. The distribution of the number of explorative pulls across the 1000 networks shows *i)* a peak at zero exploration and *ii)* a broad tail of mean explorative pulls (figure 2), suggesting that there are indeed two classes of networks. One class never pulls the stochastic arm, i.e., those networks adopt a non-learning strategy. The other class learns. For task parameters further away from the edge, this bimodality disappears.

The two behavioral regimes are characterized by distinct recurrent dynamics of the trained LSTM agents. The two left-most columns of figure 3 display the policy entropy and hidden state statistics for a network trained on a $\sigma_l, \sigma_p$-combination associated with the regime in which learning is the optimal behavior. We differentiate between the case in which the deterministic arm is the better one ($\mu < 0$) and the case in which the second arm should be preferred ($\mu > 0$). In both cases the agent first explores in order to identify the better arm. Moreover, the hidden dynamics appear to display two different attractors, which correspond to either of the arms being the better choice. The better arm can clearly be identified from the PCA-dimensionality reduced hidden state dynamics (bottom row of figure 3). The two right-most columns of figure 3, on the other hand, depict the same statistics for a network that was meta-trained on the regime in which the optimal strategy is not to learn. Indeed, the agent always chooses the deterministic arm, regardless of whether it is the better choice. Accordingly, the network dynamics seem to fall into a single attractor.

We examined how these strategies evolve over the course of meta-training and find that there are two phases: After an initial period of universal random behavior across all conditions, the distinct behavioral regimes emerge (supplementary figure 9). We note that this observation may be partially

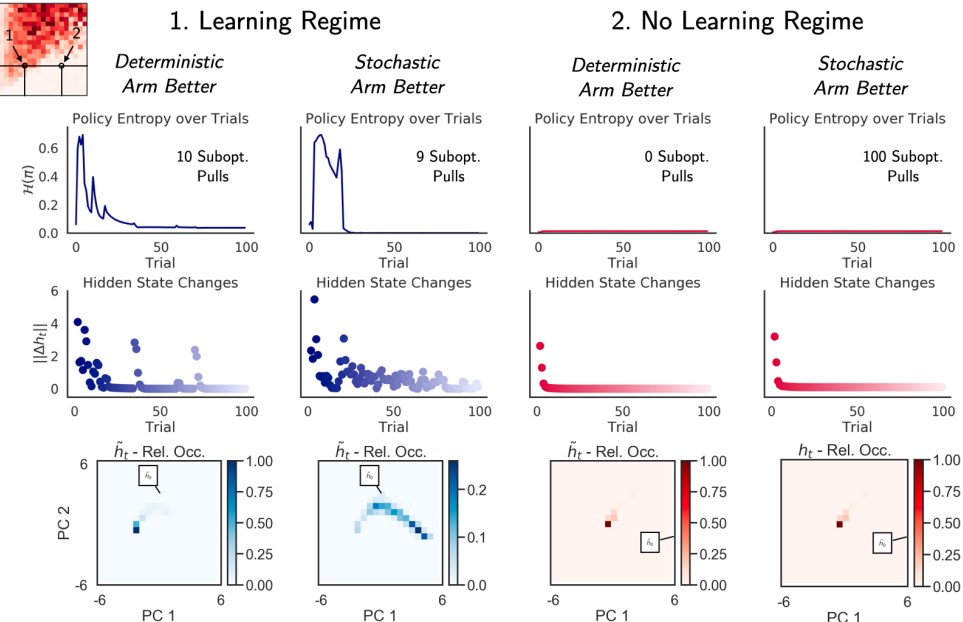

Figure 3: Recurrent dynamics of two meta-trained bandits for task conditions that favour learning (blue) and not learning (red), for a lifetime $T = 100$. Each bandit is tested both for the case where the deterministic arm is the better option and for the case where the stochastic arm is the better option. **First two columns**. Bandit for which an adaptive strategy is predicted by the theory. The inner loop dynamics integrate information which is reflected by the convergence of the hidden state to two different fixed points depending on which arm is optimal. **Last two columns**. Bandit for which the heuristic choice of the deterministic arm is the Bayes optimal behavior. Not learning manifests itself in non-explorative, rigid behavior and activations. The final row visualizes the PCA-dimensionality reduced hidden state dynamics ($\tilde{h}_t$) averaged over 100 episodes. A darker color indicates a more frequent occupancy in the discretized PCA space of the transformed hidden states.

caused by the linear annealing of the entropy regularization coefficient in the actor-critic objective which we found to be crucial in training the networks.

In summary, we observe that the meta-learned strategy shows a highly nonlinear, partially discontinuous dependence on task parameters. In transition regions between strategies, we find local maxima in the reward landscape that correspond to different learning strategies. In the simple bandit setting, these local maxima correspond to a learning and a non-learning strategy, respectively, hence providing a minimal model for a sharp nature-nurture trade-off. Next, we investigate whether these insights generalize to more complex domains by studying spatial reasoning.

## 4    TIME HORIZONS, META-LEARNED STRATEGIES & ENTROPY REDUCTION

While the simple bandit task provides an analytical perspective on the trade-off of learning versus hard-coded behavior, it is not obvious that the obtained insights generalize to more complex situations, i.e., to distributions of finite-horizon MDPs. To investigate this, we studied exploration behavior in an ensemble of grid worlds task. We hypothesize that meta-learning yields qualitatively different spatial exploration strategies depending on the lifetime of the agent. For short a lifetime, the agent should opt for small rewards that are easy to find. For longer lifetimes, the agent can spend time to explore the environment and identify higher rewards that are harder to find.

To test this hypothesis, we train a RL$^2$-based meta learner to explore a maze with three different types of goal locations (top row of figure 4): $g_h$ (green object), $g_m$ (yellow object) and $g_s$ (pink object) with transition rewards $R(g_h) > R(g_m) > R(g_s)$.

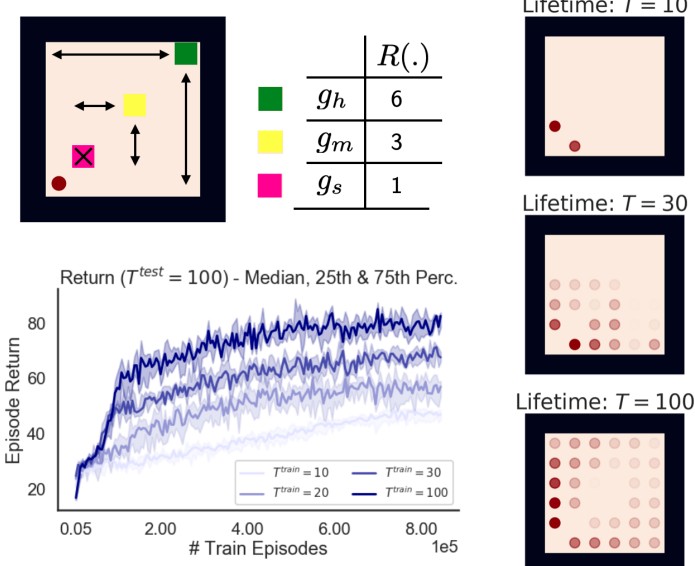

Figure 4: Grid navigation task with 3 different rewards, which differ in the amount of reward and the uncertainty in location. **Top-Left**: Task formulation. **Bottom-Left**: Learning curves ($T^{test} = 100$) for different training lifetime. We plot the median, 25th and 75th percentile of the return distribution aggregated over 10 independent training runs and 10 evaluation episodes. Agents trained on a shorter lifetimes gradually generalize worse to the larger test lifetime. **Right**: Relative state occupancy of meta-learned exploration strategies for different training lifetimes during meta-learning (averaged over 100 episodes of length 100). With increasing lifetime the agent explores larger parts of the state space and actively avoids suboptimal object location transition.

During an episode/lifetime the goal locations are fixed. At the beginning of the episode $g_m$ and $g_h$ are randomly sampled. The location of $g_s$, on the other hand, remains fixed across all training episodes. We sample the possible locations for $g_h$ from the outermost column and row (11 locations) while $g_m$ varies along the third row and column (excluding the borders, 5 locations). Thereby, the three goals encode destinations with varying degrees of spatial uncertainty and payoff. The agent can move up, down, left and right. After it (red circle) transitions into a goal location, it receives the associated reward and is teleported back to the initial location in the bottom left corner. Within one episode, the agent can hence first perform one or several exploration runs, in which it identifies the object location, and then do a series of exploitation runs, in which it takes the shortest path to that location. Importantly, the agent does not observe the goal locations but instead has to infer the locations based on the observed transition rewards.

Depending on the lifetime of the agent during the inner loop adaptation, we find that meta-learning can imprint 3 qualitatively different strategies (figure 4 right column; figure 5): For small lifetimes the agent executes a hard-coded policy that repeatedly walks to the safe, low-reward pink object. As the lifetime and consequently inner loop adaptation is increased, we find that the agent's meta-learned policy starts to explore a broader range of locations int the maze, first exploring possible locations of the medium-reward object and – for long lifetimes – the distant and uncertain high-reward object (figure 4 right column). Consistently, the agent exploits increasingly uncertain rewards with increasing lifetime (figure 5).

Furthermore, we investigated how meta-learned strategies generalize across different timescales of adaptation. More specifically, we trained an agent to learn (or not) with a given lifetime and tested how the learned behavior performed in a setting where there is more or less time available. As predicted, we find that the test time-normalized return of the agents decreased with the discrepancy between training and test lifetime (figure 6). This can be problematic in settings where the agent does not have access to its exact lifetime and highlights the lack of time-robustness of meta-adaptation.

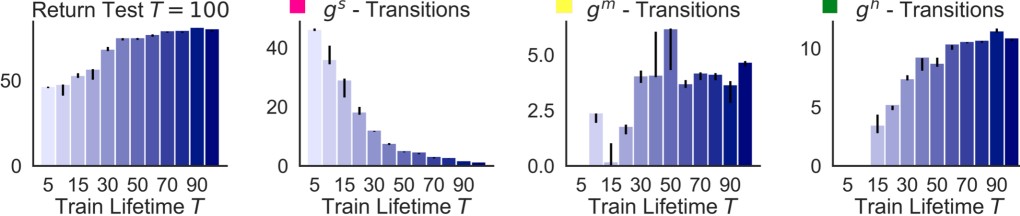

Figure 5: Lifetime dependence of the performance and the visitation counts of the goal locations for an RL$^2$ agent trained on random $6 \times 6$ grid worlds and evaluated on $T^{test} = 100$. For small lifetimes the meta-learned strategy only exploits the small safe object. For larger lifetimes the agent first explores the more uncertain medium (and later high) reward object locations. The displayed statistics (median, 25th/75th percentile) are aggregated over 5 independent training runs and 500 evaluation episodes.

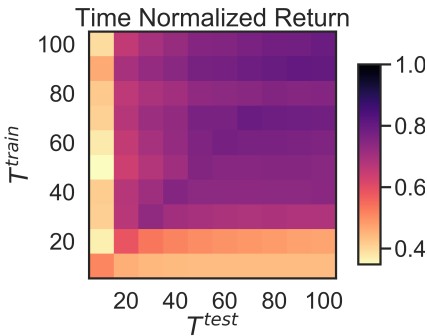

Figure 6: Episode return for agents trained on $T^{train}$ and tested on $T^{test}$. The returns are normalized by the test lifetime. Agents which were meta-trained on a fixed lifetime do not generalize well to smaller or larger test lifetimes. The statistics are averaged over 5 independent training runs and 500 test episodes.

The agents displayed clear hallmarks of model-based behavior and behavioral changes over their lifetime (figure 7). When the agent has encountered the high reward once, it resorts to a deterministic exploitation strategy that follows a shorter trajectory through the environment than the one initially used during exploration. Furthermore, the adaptive policies identify when there is not enough time left in the episode to reach the previously exploited goal location. In that case the policies switch towards the easier to reach small goal location. The oscillating policy entropy (columns three and four of figure 7) is state-specific and indicates that the meta-learned strategies have correctly learned a transition model of the relevant parts of the environment. If an action does not affect the overall length of the trajectory to a goal, this is reflected in the entropy of the policy. Finally, we analyzed the distinct recurrent dynamics for the three different strategies (final column of figure 7). We find that the dimensionality of the dynamics increases with the adaptivity of the behavior. As the training lifetime increases, the participation ratio (Gao et al., 2017) of the hidden state dynamics increases and the explained variance of the first three principal components drops.

## 5 DISCUSSION & FUTURE WORK

This work has investigated the interplay of three considerations when designing meta-task distributions: The diversity of the task distribution, task complexity and training lifetime. Depending on these, traditional meta-learning algorithms are capable of flexibly interpolating between distilling a learning algorithm and hard-coding a heuristic behavior. The different regimes emerge in the outer loop of meta-learning and are characterized by distinct recurrent dynamics shaping the hidden activity. Meta-learned strategies showed limited generalization to timescales they were not trained on, emphasizing the importance of the training lifetime in meta-learning.

A key take-home from our results is the highly nonlinear and potentially discontinuous dependence of the meta-learned strategy on the parameters of the task ensemble. For certain parameter ranges, the reward landscape of the meta-learning problem features several local maxima that correspond to different learning strategies. The relative propensity of these strategies to emerge over the course of meta-learning depends on the task parameters and on the initialization of the agent. Generally, this supports the notion that there is not a single inductive bias for a given task distribution. Rather, there could be a whole spectrum of inductive biases that are appropriate for different amounts of training data. Even for the same task setting, different training runs can result in qualitatively different

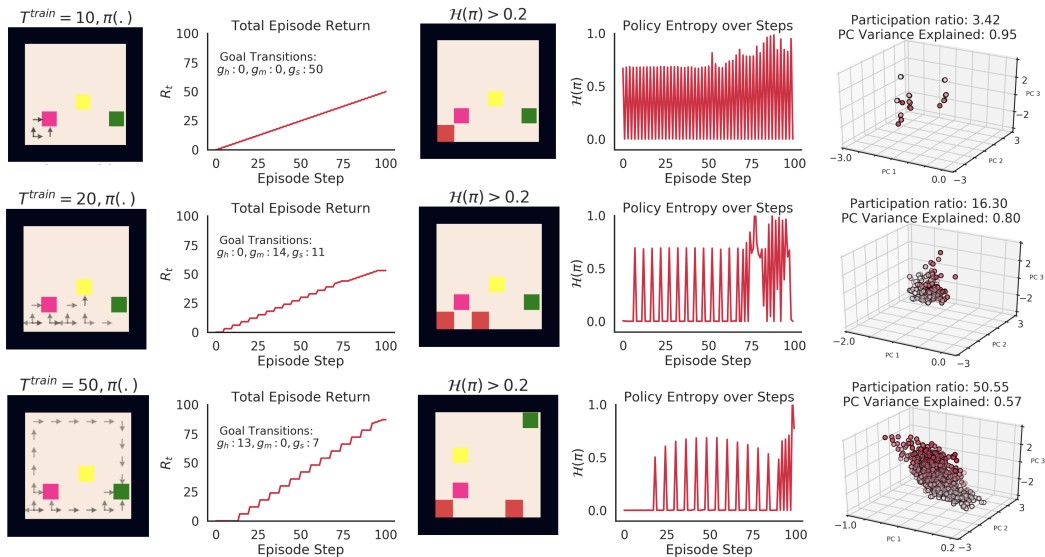

Figure 7: Characteristic trajectories for three different types of meta-learned strategies. **Top to bottom**: Episode rollouts ($T^{test} = 100$) for inner loop training lifetimes $T^{train} = \{10, 20, 50\}$. **First four columns**: The agents' trajectories, episode return, states with high average policy entropy (red squares) and the policy entropy for the same sampled environment. **Final column**. PCA-dimensionality reduced hidden state dynamics for different 100 rollout episodes and the agents' respective training lifetimes. A lighter color indicates later episode trials. With increasing lifetime the meta-learned strategies become more adaptive and the recurrent dynamics higher dimensional.

solutions, providing a note of caution for interpretations drawn by pooling over ensembles of trained networks.

The observed nonlinear dependence of the obtained solution may be relevant, e.g., for robotic applications, in which a rapid adaption of controllers trained in simulation to real-world robotic devices is desirable (e.g., Nagabandi et al., 2018; Belkhale et al., 2020; Julian et al., 2020). It is beneficial to ensure rapid adaptation on the real robot, e.g., to avoid physical damage. To achieve this, the meta-learner should be optimized for a short horizon. This, however, introduces a bias towards not learning, or in more complex settings, for heuristic solutions that explore less than is required to discover the optimal policy. For such problems, the curation of the meta-learning task ensemble may have to additionally take into account potential nonlinear and long-lasting trade-offs between final performance and speed of adaptation.

The observed sharp transition between exploratory learning behavior(s) and hard-coded, non-learning strategies can be seen as a proof-of-concept example for a "nature-nurture" trade-off that adds new aspects to earlier work in theoretical ecology (Stephens, 1991). From this perspective of animal behavior, meta-learning with a finite time horizon could provide an inroad into understanding the benefits and interactions of instinctive and adaptive behaviors. Potential applications could be the meta-learning of motor skills in biologically inspired agents (Merel et al., 2019) or instinctive avoidance reactions to colours or movements. The degree of biological realism that can be reached will be limited by computational resources, but qualitative insights could be gained, e.g., for simple instinctive behaviors. A different extension of our analysis is that to non-stationary environments, although we suspect a qualitative analogy of lifetime in our approach and auto-correlation times in non-stationary environments.

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

## A    SUPPLEMENTARY MATERIALS

### A.1    MATHEMATICAL DERIVATION OF OPTIMAL EXPLORATION

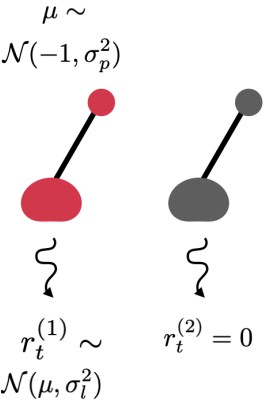

Gaussian Multi-Arm Bandit
Problem Formulation

$\mu \sim$
$\mathcal{N}(-1, \sigma_p^2)$

$r_t^{(1)} \sim$  $\quad r_t^{(2)} = 0$
$\mathcal{N}(\mu, \sigma_l^2)$

Figure 8: Two-arm Gaussian bandit.

In the following section we describe the Gaussian two-arm bandit setting analyzed in section 3. The first arm generates stochastic rewards $r$ according to hierarchical Gaussian emissions. Between individual episodes the mean reward $\mu$ is sampled from a Gaussian distribution with standard deviation $\sigma_p$. The second arm, on the other hand, arm generates a deterministic reward of 0. More specifically, the generative process for rewards resulting from a pull of the first arm is described as follows:

$$\mu \sim \mathcal{N}(-1, \sigma_p^2); \ r \sim \mathcal{N}(\mu, \sigma_l^2)$$

$$p(\mu | \sigma_p^2) = \frac{1}{\sqrt{2\pi\sigma_p^2}} \exp\left\{ -\frac{(\mu + 1)^2}{2\sigma_p^2} \right\}$$

$$p(r | \mu, \sigma_l^2) = \frac{1}{\sqrt{2\pi\sigma_l^2}} \exp\left\{ -\frac{(r - \mu)^2}{2\sigma_l^2} \right\} \ .$$

Our Bayesian agent is assumed to spend a fixed amount of trials $n$ of the overall lifetime $T$ exploring the second stochastic arm. This assumption is justified since our corresponding meta-learning agent may easily encode the deterministic nature of the fixed arm 0 and therefore only explore the second non-deterministic arm.[2] The expected cumulative reward of such a two-phase exploration-exploitation policy can then be factorized as follows:

$$\mathbb{E}_{\mu,r}\left[ \sum_{t=1}^{T} r_t \, \bigg| \, n \right] = \mathbb{E}_{\mu,r}\left[ \sum_{t=1}^{n} r_t \, \bigg| \, n \right] + \mathbb{E}_{\mu,r}\left[ \sum_{t=n+1}^{T} r_t \, \bigg| \, n, \hat{\mu} \right]$$

$$= (-1) \times n + \mathbb{E}_\mu \left[ (T - n) p(\hat{\mu} > 0) \mu \right]$$

where $\hat{\mu}$ denotes the maximum a posteriori (MAP) estimate of $\mu$ after n trials:

$$\hat{\mu} = \frac{1}{P_{tot}} \left[ -1 \times P_p + n \times P_l \times \bar{r} \right] \ .$$

with $P_p = \frac{1}{\sigma_p^2}$, $P_l = \frac{1}{\sigma_l^2}$, $P_{tot} = P_p + nP_l$ and $\bar{r} = \frac{1}{n}\sum_{i=1}^{n} r_i$. For a fixed $\mu$ (within a lifetime) this random variable follows a univariate Gaussian distribution with sufficient statistics given by:

$$\mathbb{E}[\hat{\mu}] = \frac{1}{P_{tot}} \cdot (-P_p + nP_l \bar{r}) \, ; Var[\hat{\mu}] = \frac{1}{P_p + nP_l} \ .$$

The probability of exploiting arm 2 after $n$ exploration trials is then given by the following integral:

$$p(\hat{\mu} > 0) = \int_0^\infty p(\hat{\mu}) d\bar{\mu} \ .$$

$\mathbb{E}_{\mu,r}[\sum_{t=1}^{T} r_t | n]$ may then be evaluated by numerical integration and the resulting optimal $n^\star$ is obtained by searching over a range of $n = 1, \ldots, T$.

---

[2]The second column of figure 3 validates this assumption since the policy entropy quickly vanishes after an initial exploration phase.

## A.2 EXPERIMENTAL DETAILS

### A.2.1 MEMORY-BASED META-REINFORCEMENT LEARNING

We follow the standard $RL^2$ paradigm (Wang et al., 2016) and train an LSTM-based actor-critic architecture using the A2C objective (Mnih et al., 2016):

$$\mathcal{L}^{AC} = \mathcal{L}^{\pi} + \beta_v \mathcal{L}^v - \beta_e \mathcal{L}^e$$
$$\mathcal{L}^{\pi} = \mathbb{E}_{\pi}\left[\log \pi(a_t|x_t)[R_t - V(x_t)]\right]$$
$$\mathcal{L}^v = \mathbb{E}_{\pi}\left[(R_t - V(x_t))^2\right]$$
$$\mathcal{L}^e = \mathbb{E}_{\pi}\left[\mathcal{H}(\pi(a_t|x_t))\right]$$
$$R_t = \sum_{i=0}^{T-t-1} \gamma^i r_{t+i},$$

where $R_t$ denotes the cumulative discounted reward resulting from the rollout of the episode. The agent interacts with a sampled environment for a single episode. Between episodes a new environment is sampled. Unless otherwise stated we ensure a proper scaling of the timestamp input and follow Pardo et al. (2017) by normalizing the time input to lie between -1 and 1.

### A.2.2 GAUSSIAN MULTI-ARM BANDITS: HYPERPARAMETERS

All results of section 3 for the two-arm Gaussian bandit setting (and all $\sigma_l$, $\sigma_p$-combinations) may be reproduced using the following set of hyperparameters:

| Parameter | Value | Parameter | Value | Parameter | Value |
|---|---|---|---|---|---|
| Training episodes | 30k | Learning rate | 0.001 | $L_2$ Weight decay $\lambda$ | $3e-06$ |
| Clipped gradient norm | 10 | Optimizer | Adam | Workers | 2 |
| $\gamma_T$ | 0.999 | $\beta_{e,T}$ | 0.005 | $\beta_v$ | 0.05 |
| $\gamma_0$ | 0.4 | $\beta_{e,0}$ | 1 | LSTM hidden units | 48 |
| $\gamma$ Anneal time | 27k Ep. | $\beta_e$ Anneal time | 30k Ep. | Learned hidden init. | ✓ |
| $\gamma$ Schedule | Exponential | $\beta_e$ Schedule | Linear | Forget gate bias init. | 1 |
| - | - | - | - | Orthogonal weight init. | ✓ |

Table 1: Hyperparameters (architecture & training procedure) of the bandit A2C agent.

We use the same set of hyperparameters for all $\sigma_l$, $\sigma_p$ and $T$ combinations. Our theoretical results are derived for the case of $\gamma = 1$. This poses a challenge when training recurrent policies. The difficulty of the temporal credit assignment problem is implicitly defined by the effective length of the time window. We observed that starting with a large $\gamma$ often times hindered the network in learning. Hence, we discount factor as a form of implicit curriculum and annealed it accordingly.

### A.2.3 GRIDWORLD NAVIGATION TASK: HYPERPARAMETERS

| Parameter | Value | Parameter | Value | Parameter | Value |
|---|---|---|---|---|---|
| Training episodes | 1M | Learning rate | 0.001 | $L_2$ Weight decay $\lambda$ | 0 |
| Clipped gradient norm | 10 | Optimizer | Adam | Workers | 7 |
| $\gamma$ | 0.99 | $\beta_{e,T}$ | 0.5 | $\beta_v$ | 0.1 |
| $\beta_e$ Schedule | Linear | $\beta_{e,0}$ | 0.01 | LSTM hidden units | 256 |
| - | - | $\beta_e$ Anneal time | 700k | Learned hidden init. | ✓ |

Table 2: Hyperparameters (architecture & training procedure) of the gridworld A2C agent.

In order to train the agents that generated the state occupancies in figure 4 and the rollout trajectories in figure 7 we additionally annealed the discount factor starting at 0.8 to 1 within the first 800k episodes.

## A.3 SUPPLEMENTARY FIGURES

Figure 9: Learning to learn and not to learn. **Left to right**. The meta-learned exploration strategy is visualized for different increasing checkpoints throughout meta-learning ($T = 100$). The initial behavior is random, but as training progresses two distinct regimes (adaptive versus heuristic) emerge. The amount of meta-learned exploration is averaged both over 5 independent training runs and 100 episodes for each of the 400 trained networks.

