# OpenReview forum: "Learning not to learn: Nature versus nurture in silico"
_ICLR.cc/2021/Conference — Reject_

### Official Review · AnonReviewer4 · 2020-10-28

**Rating:** 5
**Confidence:** 4

**Review:**

Summary: This paper observes that in meta-RL (and evolutionary biology), sometimes it is advantageous to learn behaviors that adapt to the particular task, while other times not adapting to the task, and instead relying on a task-agnostic “hard-coded” behavior is sufficient. While much meta-RL research typically focuses on the former setting, this paper studies when it is not necessary to learn adaptive behaviors. Specifically, this paper presents three main findings: (i) whether or not it is optimal to learn adaptive behavior strongly depends on the horizon of the task and complexity of learning such adaptive behaviors — if the horizon is too short, or if the adaptive behavior requires complex exploration, then exploring the new task to learn adaptive behaviors may not be worth it; (ii) existing meta-RL agents are capable of choosing not to learn adaptive behaviors, when it is optimal to do so; (iii) existing meta-RL agents generalize poorly to tasks with varying horizon-lengths.

Strengths:
- Novelty. Most meta-RL papers study the case where adapting to a new task requires learning new behaviors. Instead, this paper studies when such adaptation is not necessary, which is an area that has not been well-studied. Therefore, it is interesting and novel to bring attention to the fact that both regimes exist.
- Clarity and Execution. This paper is generally well-executed and clear. The bandit example in Section 3 clearly illustrates the two regimes that this paper studies: when it is necessary to learn adaptive behaviors vs. not, and the results are presented in figures that impressively, clearly illustrate the points made in the text. This section also convincingly shows that existing meta-RL agents roughly learn the Bayes-optimal policy in this case. Similarly, the grid world tasks in Section 4 also clearly illustrate how the behavior of meta-RL agents changes with the horizon of the task. The paper does a good job at thoroughly studying what happens when each parameter (e.g., horizon-length, aleatoric uncertainty, and epistemic uncertainty) varies.

Weaknesses:
- Significance. My primary concern with this work is significance. I believe that the main insights of this paper can be interpreted through the lens of Bayes-adaptive policies [1]. The optimal Bayes-adaptive policy explains when it is optimal to explore a new task and learn adaptive behaviors, depending on the horizon length and amount of exploration needed, and therefore, the main claim of the paper can be framed as observing that existing meta-RL agents can learn the Bayes-adaptive optimal policy in simple tasks. This is an interesting observation, but prior work [2] already shows that meta-RL is equivalent to learning the Bayes-adaptive optimal policy. Furthermore, for harder tasks, this is likely to be confounded with the complexity of learning efficient exploration behaviors that allow learning adaptive behaviors, as shown by prior work [2, 3, 4]. Overall, it is therefore unclear to me what significant takeaways the meta-RL community can gain from this work. One potential takeaway is the observation that meta-RL agents generalize poorly to tasks with varying horizon-lengths, but it’s unclear whether this setting occurs in real tasks, and even if this _is_ necessary, then simply adding the horizon to the observation (state) and varying the horizon during meta-training seems sufficient.

Overall, I found this paper to be a well-written, well-executed, illustration of when it is optimal to learn task-specific adaptive behaviors. However, due to its limited significance, I am unfortunately unable to recommend acceptance.

Additional Questions / Comments:
- In Figure 2, what does it mean with “100 Suboptimal pulls?” Is this over the course of all training? Also, why is the initial policy entropy at 0? I would expect that upon initialization, the policy entropy is not 0.
- I believe that the Bayes-adaptive optimal policy can also be analytically computed for the grid world tasks. It would be nice to report the optimal returns in Figure 5.
- The interplay between the term “lifetime” and episodes was not explicitly defined. It was possible to infer the paper’s intent from reading the experiments, but more carefully defining this could help.
- Framing the exploration trade-off in terms of evolutionary biology is interesting.

References:

[1] Optimal Learning: Computational procedures for Bayes-adaptive Markov decision processes. Michael O’Gordon Duff. February 2002. https://www.gatsby.ucl.ac.uk/~yael/Okinawa/DuffThesis.pdf

[2] VariBAD: A Very Good Method for Bayes-Adaptive Deep RL via Meta-Learning. Luisa Zintgraf, Kyriacos Shiarlis, Maximilian Igl, Sebastian Schulze, Yarin Gal, Katja Hofmann, Shimon Whiteson. October 2019. https://arxiv.org/abs/1910.08348

[3] Efficient Off-Policy Meta-Reinforcement Learning via Probabilistic Context Variables. Kate Rakelly, Aurick Zhou, Deirdre Quillen, Chelsea Finn, Sergey Levine. March 2019. https://arxiv.org/abs/1903.08254

[4] Explore then Execute: Adapting without Rewards via Factorized Meta-Reinforcement Learning. Evan Zheran Liu, Aditi Raghunathan, Percy Liang, Chelsea Finn. June 2020. https://openreview.net/forum?id=La1QuucFt8-

------------

**Edit: Score raised from 4 --> 5 following discussion below.**

---

> ### Author Response · Authors · 2020-11-18
> **Reply to Review by AnonReviewer4 - Part 1**
>
> Thank you very much for your clear comments. We would like to first take the opportunity to address the key question of significance.
>
> **[... therefore, the main claim of the paper can be framed as observing that existing meta-RL agents can learn the Bayes-adaptive optimal policy in simple tasks. This is an interesting observation, but prior work [2] already shows that meta-RL is equivalent to learning the Bayes-adaptive optimal policy. ] "Overall, it is therefore unclear to me what significant takeaways the meta-RL community can gain from this work. One potential takeaway is the observation that meta-RL agents generalize poorly to tasks with varying horizon-lengths ..."**
>
> We agree that the meta-learner effectively learns the Bayes-adaptive optimal policy, but we do not think that this is our main take-home message. On a general level, meta-learning can probably always be phrased as an algorithmic way of inferring a Bayes-optimal policy.  Aside from suggesting meta-learning approaches as a principled approach to nature-nurture problems in evolutionary biology, this work aims at mapping out the dependence of the optimal policy on the task parameters. We believe that the meta-learning community has paid little attention to the effects of the meta-training distribution and the amount of the inner loop adaptation. The main take home is that these effects can be nonlinear to the point of discontinuity: The switch from the learning to the non-learning regime is a sharp one. The underlying reason is local maxima in the reward landscape (evidence for that in Fig. 2 of the revised manuscript). These types of discontinuity (in seemingly unimportant parameters like lifetime) is interesting not only from the perspective of evolutionary biology -- as sharp nature-nurture transitions -- but  may well occur in a range of other meta-learning tasks and could have a strong impact on gradient-based meta-learning approaches.
>
> We try to better highlight these points in the revised version. To this, we added an additional analysis close to the transition edge to illustrate the presence and impact of these local maxima (Fig. 2). We also added a paragraph to the discussion:
>
> *"[...] A key take-home from our results is the highly nonlinear and potentially discontinuous dependence of the meta-learned strategy on the parameters of the task ensemble. For certain parameter ranges, the reward landscape of the meta-learning problem features several local maxima that correspond to different learning strategies. The relative propensity of these strategies to emerge over the course of meta-learning depends on the task parameters and on the initialization of the agent. Generally, this supports the notion that there is not a single inductive bias for a given task distribution. Rather, there could be a whole spectrum of inductive biases that are appropriate for different amounts of training data. Even for the same task setting, different training runs can result in qualitatively different solutions, providing a note of caution for interpretations drawn by pooling over ensembles of trained networks."*
>
> **"Furthermore, for harder tasks, this is likely to be confounded with the complexity of learning efficient exploration behaviors that allow learning adaptive behaviors, as shown by prior work [2, 3, 4]"**
>
> Yes, limited generalization can of course occur for a variety of reasons, but isn't it interesting to know that there could be a deeper underlying reason why things don't work as well as hoped for? We found it interesting, for example, that the washed-out edge between learning and non-learning in the simulated RL2 agents is in fact not due to a gradual change in the policy, but due to changing probabilities of falling into totally different policies (see newly added figure 2).
>
> **"[...] even if this is necessary, then simply adding the horizon to the observation (state) and varying the horizon during meta-training seems sufficient."**
>
> That feels like a natural extension, but given that potentially very nonlinear dependencies of the optimal policy on the horizon, it may not be so easy to train. It would certainly be worth investigating such a horizon-dependent meta-learner, and be it to allow the use of the horizon as a gradually increasing curricular parameter and thereby provide a meta-learner that adapts very quickly on the job, without hampering its precision as more training data comes in.

---

> > ### Author Response · Authors · 2020-11-18
> > **Reply to Review by AnonReviewer4 - Part 2**
> >
> > **"In Figure 2, what does it mean with “100 Suboptimal pulls?” Is this over the course of all training?"**
> >
> > We modified the caption to hopefully better explain the figure setup:  *Recurrent dynamics of two meta-trained bandits for task conditions that favour learning (blue) and not learning (red), for a lifetime $T=100$. Each bandit is tested both for the case where the deterministic arm is the better option and for the case where the stochastic arm is the better option.*
> >
> > **"Also, why is the initial policy entropy at 0? I would expect that upon initialization, the policy entropy is not 0."**
> >
> > We optimize not only the weights of the LSTM, but also its initial condition. Therefore, the network can start with a configuration in which the entropy is zero. This is stated in the methods section.
> >
> > **"[...] I believe that the Bayes-adaptive optimal policy can also be analytically computed for the grid world tasks."**
> >
> > We are curious about how to analytically compute the optimal policy in closed-form for the gridworld problem? We attempted this multiple times but didn’t succeed. This was due to the exponential branching of possible entropy reduction trajectories as well as the lack of a unique optimal policy. Zintgraf et al. (2020) use an arguably simpler gridworld formulation and also didn’t provide an analytical baseline for the optimal BAMDP solution. Could you provide further intuition?
> >
> > The interplay between the term “lifetime” and episodes was not explicitly defined. It was possible to infer the paper’s intent from reading the experiments, but more carefully defining this could help.
> >
> > We overhauled the arguably bit convoluted description of the bandit task and hope it's clearer now:
> >
> > *"The agent experiences episodes consisting of T arm pulls, representing the lifetime of the agent. The statistics of the bandit are constant during each episode, but vary between episodes."*

---

> > > ### Comment · AnonReviewer4 · 2020-11-20
> > > **Further clarification about the main claims**
> > >
> > > Thanks for the detailed response!
> > >
> > > **Main claims of the paper:** I admit that I don't fully understand the above framing about the paper's results, and I'd love to get some additional clarification here. To reiterate, my main concern is that the paper's results are explained by the Bayes-optimal policy: the observed discontinuity between the learning and not-learning regimes occurs for Bayes-optimal policies as uncertainty levels / horizon lengths change.
> > >
> > > If I'm understanding this response correctly, this is attempting to frame the paper along the lines of: "while the effects of varying uncertainty / horizon lengths is well-understood for Bayes-optimal policies, it is not understood for existing meta-RL approaches, which is the topic of this paper"? I agree that this could be an interesting direction, although in this case, I think carefully handling the exploration in meta-RL problem is necessary, e.g., via more sophisticated approaches like [2, 3, 4] above, to avoid confounding.
> > >
> > > In any case, my review mainly hinges on this point, and I'd be receptive to clarification from the authors.
> > >
> > > **Bayes-adaptive optimal policy for grid world tasks:** Is it true that for any time horizon $T$, the Bayes-adaptive optimal policy is only one of three behaviors: (i) always go to the green block, (ii) always go to the yellow block, (iii) always go to the pink block, with only minor mixing (i.e., in case (i), the last few time steps may go to the pink block, if there are not enough time steps to go to the green block anymore)? I think some variant of this is true, which would make computing the optimal returns straightforward.

---

> > > > ### Author Response · Authors · 2020-11-23
> > > > **Reply II to  AnonReviewer4 - Part 1**
> > > >
> > > > Thank you very much for the quick response and the opportunity for clarification.
> > > >
> > > > **“[...]  ‘while the effects of varying uncertainty / horizon lengths is well-understood for Bayes-optimal policies, it is not understood for existing meta-RL approaches, which is the topic of this paper’?”**
> > > >
> > > > Yes, the above highlighted insight is one key result of our analysis. A related contribution is our observation that there exists a bimodal solution distribution directly at the border between learning and non-learning regime (see figure 2). We show that the inner-loop policy can either be meta-learned to implement an adaptive and explorative learning strategy or learn the deterministic behavior depending on the initialization of the RNN. We validate this insight on 1000 seeds for the bandit case. This points to an unwanted failure of the analogy between memory-based meta-learning and optimal Bayesian inference: The solution space can contain local minima, which correspond to very different behaviors and are hard to escape with first-order optimization methods. Furthermore, we have found indications of the same phenomenon in the gridworld task (see new page 2 of [Google doc animations](https://docs.google.com/document/d/1bnmIykdsOase4QPgh4LkH-LtuDczgRPbiKRSM3M42jc/edit?usp=sharing)) and are currently in the process of validating this on an extended number of seeds. Unfortunately, the simulations take too long to be able to provide you with conclusive results until the end of this discussion, but we would be happy to include these results in a potential camera-ready version of the paper.
> > > >
> > > >
> > > > **“[...] I agree that this could be an interesting direction, although in this case, I think carefully handling the exploration in meta-RL problem is necessary, e.g., via more sophisticated approaches like [2, 3, 4] above, to avoid confounding.”**
> > > >
> > > > Thanks for the suggestion. While the suggested approaches are certainly more sophisticated than RL^2, we believe that not all of them are straightforward to apply to our setting, and that their use would probably not change the final results. Here's a short discussion how the cited work relates to our contributions, we'd be happy to include a shortened version in the manuscript if you think it improves the paper:
> > > >
> > > > - PEARL (Rakelly et al., 2019, [3]) infers task-context variables using a feedforward architecture and the SAC algorithm (Haarnoja et al., 2018). It is not a memory-based meta-learning algorithm in its proposed form. Instead it explicitly “outsources” the task inference to an encoder architecture (similar to Zintgraf et al., 2018) and requires multiple episodes for context inference. We believe that this constitutes an interesting idea, but do not think that it solves the fundamental optimization/credit assignment problem at the edge between the two regimes. At this point in the task distribution space, the task inference is especially challenging due to the negative interference of the gradient signals. Even if sufficient exploration is conducted in a particular episode, this will be suboptimal for certain sampled tasks.
> > > >
> > > >
> > > > - DREAM (Liu et al., 2020, [4]) extends upon Rakelly et al. (2019, [3]) by allowing the task encoder to infer a context variable without the need for any reward signal. This does not apply to our bandit setting, since here the reward is fundamental to infer the task characteristics. Furthermore, DREAM requires an explicit ‘exploration’ episode and multiple following exploitation episodes. Hence, the task formulation is challenging to compare to our setting.
> > > > Finally, the learning curve results shown in figures 3 and 6 of Liu et al. (2020, [4]) indicate that DREAM does not achieve qualitatively different policies from RL^2, but does so in a more sample efficient way. Importantly, we do not focus on improving sample efficiency and train the RL^2 for an extensive period of time 30k episodes (as compared to 20k in Wang et al., 2016) in the bandit setting and only evaluate the final converged solution.

---

> > > > > ### Author Response · Authors · 2020-11-23
> > > > > **Reply II to AnonReviewer4 - Part 2**
> > > > >
> > > > > - We relate VariBAD (Zintgraf et al., 2019, [2]) to our work in section 2 of the manuscript. Similar to DREAM, their setup deals with meta-learning adaptation strategies across multiple episodes. In order to analyze the impact of the agent’s time horizon, we focus on the case of a single episode and within-episode adaptation. Furthermore, in the bandit setting we do not only look at a single task distribution, but at the solution space across the task distribution space. Finally, in follow-up work, Zintgraf et al. (2020) show that variBAD requires further exploration bonuses akin to random network distillation (Burda et al., 2018) in hyper-state space. Hence, we do not believe that standard VariBAD on its own is capable of solving the discovered bimodality (figure 2) of the meta-learned solutions. But we agree that a solution to our observation may require rethinking of meta-learning objective functions and provides an interesting avenue for future work.
> > > > > In summary, the above mentioned methods provide significant contributions to increasing the efficiency of meta-reinforcement learning algorithms and a selected subset of task distributions. Our work, on the other hand, is interested in understanding the underlying principles of memory-based meta-learning across the entire space of task distributions and potentially interfering gradient signals.
> > > > >
> > > > > **“[...] (i) always go to the green block, (ii) always go to the yellow block, (iii) always go to the pink block, with only minor mixing [...]? I think some variant of this is true, which would make computing the optimal returns straightforward.”**
> > > > >
> > > > >
> > > > > The observed behaviour indeed looks like the mentioned three policies (depending on the amount of time), but it is not obvious how to obtain such a policy in fully tractable analytical fashion. But if it strengthens the paper, we are happy to provide a heuristic baseline and the expected return, which implements these behaviors based on the available test time.
> > > > >
> > > > >
> > > > > **Additional References:**
> > > > >
> > > > >
> > > > > Burda, Y., Edwards, H., Storkey, A., & Klimov, O. (2018). Exploration by random network distillation. arXiv preprint arXiv:1810.12894.
> > > > > Haarnoja, T., Zhou, A., Abbeel, P., & Levine, S. (2018). Soft actor-critic: Off-policy maximum entropy deep reinforcement learning with a stochastic actor. arXiv preprint arXiv:1801.01290.
> > > > > Zintgraf, L., Feng, L., Igl, M., Hartikainen, K., Hofmann, K., & Whiteson, S. (2020). Exploration in Approximate Hyper-State Space for Meta Reinforcement Learning. arXiv preprint arXiv:2010.01062.
> > > > > Zintgraf, L., Shiarli, K., Kurin, V., Hofmann, K., & Whiteson, S. (2019, May). Fast context adaptation via meta-learning. In International Conference on Machine Learning (pp. 7693-7702). PMLR.

---

> > > > > > ### Comment · AnonReviewer4 · 2020-11-23
> > > > > > **Well-executed paper; Remaining concerns about signficance**
> > > > > >
> > > > > > Overall, I find this paper well-executed and quite clear, which other reviewers also seem to agree with. I appreciate the author responses, which by reframing the results, have partially alleviated my main concern about whether the paper's results can be explained exclusively via examining Bayes-adaptive optimal policies. I'm willing to raise my score to a 5.
> > > > > >
> > > > > > Despite this reframing, I still think that this direction could be more thoroughly explored. Concretely, the authors write:
> > > > > >
> > > > > > > The solution space can contain local minima, which correspond to very different behaviors and are hard to escape with first-order optimization methods.
> > > > > >
> > > > > > This hints at the exploration problem that prior works [2, 3, 4] describe, so it's unclear to me if this result indicates a broad challenge for all meta-RL approaches, or if this is just a failure mode exclusive to RL^2. Ideally, this would be tested by evaluating additional meta-RL approaches, e.g. [2, 3, 4], but if this is impossible, as the authors indicate, then at least discussing them seems useful. In particular, the authors hypothesize that VariBAD alone may be unable to solve the task, which seems reasonable, but unclear without further experimentation.

---

### Official Review · AnonReviewer1 · 2020-10-28
**Creative proposal; analysis seems flawed**

**Rating:** 5
**Confidence:** 4

**Review:**

Summary:
This paper explores the effect of time horizon on meta-reinforcement learning agents. Using a recurrent meta-learner, it demonstrates that different strategies are learned based on the time horizon during meta-training.

Pros:
Creative proposal to study when incorporating new data to adapt to a task is warranted versus executing an existing behavior.
Results are very nicely presented and writing is good
Cons:
The analogy to “nature versus nurture” seems tenuous
Results seem somewhat obvious (see comments below)

Detailed Comments:

While I appreciate the inspiration of the biological connections argued in this work, I am not convinced that the analogy of learning versus adapting in meta-learning maps neatly onto the concepts of nature versus nurture in biology. If biology terminology is going to be used, it needs to be carefully defined for a machine learning audience, and the limitations of the analogy discussed. Presently, I find that the use of this terminology subtracts from the clarity of the work.

My main criticism of the work is that the results seem fairly obvious. In the finite horizon case, the meta-learner will necessarily learn a strategy optimal for the given horizon because that’s exactly what it’s optimized for. It seems akin to me to training an RNN policy in an MDP and then noticing that the RNN is capable of *not* persisting information across timesteps since it isn’t needed. Since feedforward models are a subset of recurrent models, this seems obvious.

It also seems strange to me that the agent is only tested in 1 episode. It seems like if the agent has learned a new skill, it should be able to repeat that skill if given the chance (e.g., with a reset), and that that is a hallmark of meta-learning. What is happening in this paper seems more like “horizon-dependent RL.” I don’t really see how the behavior in the long-time horizon setting can be called “learning” while the short time-horizon behavior is not. Both seem like the optimal policy for the given MDP. In Section 4, for example, wouldn’t “learning” be defined as figuring out where the colored squares are? Yet in the paper it seems to be defined as executing a policy that moves farther away from the start. It seems to me like exploration and learning are being conflated here.

Several times throughout the paper it is claimed that the work “investigates the interplay of  three considerations when designing meta-task distributions: The diversity of the task distribution, task complexity and training lifetime.” I don’t see how the first two are analyzed in this paper. Perhaps in the bandit example, but there it seems to be always entangled with the training lifetime.

Besides the comments about the biological terminology mentioned earlier, the paper is clear and easy to read, with the other exception of the description of the 2-arm Gaussian bandit in Section 3. This reviewer had to read that first paragraph about 5 times to understand the setup. The figures are well-done and clearly present the results.

Recommendation:
5. While experiments are thorough and nicely presented, I presently don’t see what insights are gained from the analysis. The claim of analyzing “diversity of the task distribution, task complexity and training lifetime” seems like an over-claim to me. It’s not clear to me that the agent “learns” a new skill, since there is only exploration, and no exploitation.

----Update----
After reading all the other reviews and ensuing discussions, I maintain my original score. If the research question is "How does the optimal policy depend on task parameters such as uncertainty and horizon?" I believe Bayes-adaptive work answers that question. If the question is "How do policies learned by meta-RL algorithms compare to Bayes-optimal policies?" then I think more empiricism is needed (since RL2 in principle can represent the Bayes-optimal policy), or a comparison of multiple methods.

---

> ### Author Response · Authors · 2020-11-18
> **Reply to Review by AnonReviewer1**
>
> Thank you for your detailed and constructive comments. We adapted the manuscript and hope that they are addressed appropriately:
>
> **"[...]  I am not convinced that the analogy of learning versus adapting in meta-learning maps neatly onto the concepts of nature versus nurture in biology."**
>
> We do not aim to capture every dimension of the biological ‘nature versus nurture’ debate and believe that this would require an unpractical amount of complexity. We have added a discussion section to clarify our claims and to relate our results to the related work in computational ethology:
>
> *"[...] The observed sharp transition between exploratory learning behavior(s) and hard-coded, non-learning strategies can be seen as a proof-of-concept example for a "nature-nurture" trade-off that adds new aspects to earlier work in theoretical ecology (Stephens, 1991). From this perspective of animal behavior, meta-learning with a finite time horizon could provide an inroad into understanding the benefits and interactions of instinctive and adaptive behaviors. Potential applications could be the meta-learning of motor skills in biologically inspired agents (Merel et al., 2019) or instinctive avoidance reactions to colours or movements. The degree of biological realism that can be reached will be limited by computational resources, but qualitative insights could be gained, e.g., for simple instinctive behaviors." (p. 9)*
>
> **"[...] My main criticism of the work is that the results seem fairly obvious."**
>
> We agree that post-hoc, the key results feel somewhat obvious, but nevertheless believe that they have consequences not only by providing a potential new workhorse for theoretical biology, but also for a broad range of meta-learning problems. The highly nonlinear, discontinuous dependence of the optimal solution on task parameters and lifetime -- i.e., the amount of data available to the optimized learner on the job -- is a consequence of local maxima in the reward landscape. We now investigate this in more depth in figure 2, which investigates the sensitivity of the bimodal solution space directly at the edge between “learning” and “not learning” regimes. We summarized the results of this new analysis in the following paragraph in the discussion:
>
> *"[...] A key take-home from our results is the highly nonlinear and potentially discontinuous dependence of the meta-learned strategy on the parameters of the task ensemble. For certain parameter ranges, the reward landscape of the meta-learning problem features several local maxima that correspond to different learning strategies. The relative propensity of these strategies to emerge over the course of meta-learning depends on the task parameters and on the initialization of the agent. Generally, this supports the notion that there is not a single inductive bias for a given task distribution. Rather, there could be a whole spectrum of inductive biases that are appropriate for different amounts of training data. Even for the same task setting, different training runs can result in qualitatively different solutions, providing a note of caution for interpretations drawn by pooling over ensembles of trained networks." (p. 8)*
>
> **"[...] Wouldn’t “learning” be defined as figuring out where the colored squares are? Yet in the paper it seems to be defined as executing a policy that moves farther away from the start.:**
>
> We believe that there may be a misunderstanding: The object locations vary between episodes/’lives’ and the optimized agent indeed first explores the environment & identifies a reward location. It then repeatedly goes to that object (if this is optimal given the remaining lifetime). The rollouts in figure 7 only visualizes a single sampled MDP for three different agents with different training lifetimes. We provide additional episode rollout visualizations in an anonymous [Google document](https://docs.google.com/document/d/1bnmIykdsOase4QPgh4LkH-LtuDczgRPbiKRSM3M42jc/edit?usp=sharing). The agents use different strategies depending on their training lifetime and acquired inner-loop learning algorithm.
>
> **"[...] This reviewer had to read that first paragraph about 5 times to understand the setup."**
>
> We excuse the inconvenience of the convoluted bandit task description and have restructured the section to provide a ‘smoother’ reader experience. Here is the updated and hopefully better to read excerpt:
>
> *"[...] To keep it simple, one of the two arms is deterministic and always returns a reward of 0. The task distribution is represented by the variable expected reward of the other arm, which is sampled at the beginning of an episode, from a Gaussian distribution with mean -1 and standard deviation σ_p, i.e. µ ~ N(-1, σ_p^2). The standard deviation σ_p controls the uncertainty of the ecological niche. For σ_p<<1, the deterministic arm is almost always the better option. For σ_p >> 1, the chances of either arm being the best in the given episode is largely even. [...]" (p. 3)*

---

> > ### Comment · AnonReviewer1 · 2020-11-23
> > **Still confused about significance of results**
> >
> > I apologize for the late response!
> > I think that based on your response and the rollouts visualized in the Google doc, a lifetime is in fact defined to include multiple episodes? That makes sense to me in the context of meta-learning. It would be good then to correct this line in the paper (Section 2) that says, ”While their adaptation setup extends over multiple episodes, we focus on single lifetime adaptation.”
> >
> > My main concern is still the significance of the results.
> >
> > I don’t think I understand the value of the added Figure 2 and bimodal reward distribution near the boundary between learning and not-learning. It seems completely expected that if the value of “learning” or “not learning” is the same, then the agent is free to pick one strategy or the other. While the inductive bias matters at the boundary, it doesn’t matter farther from the boundary, right? So in that sense, it seems like the inductive bias doesn’t really matter at all, since in all cases we can acquire an optimal strategy. I think the idea is that in the approximate case, we might encounter local maximums that are not actually optimal (?) If this is the case, I think this problem needs to be demonstrated empirically to ensure it exists.
> >
> > Can the authors explain the logic in this paragraph? “The observed nonlinear dependence of the obtained solution may be relevant, e.g., for robotic applications...the curation of the meta-learning task ensemble may have to take into account potential nonlinear and long-lasting trade-offs between final performance and speed of adaptation: Agents that adapt quickly may stop to learn prematurely.” The first part of this paragraph is I think referencing the fact that different solutions are possible at the learning/ non-learning boundary. The second half seems to be referencing a setting where an agent is trained for short-horizon and thus adapts quickly, but then is tested in a longer-horizon setting where it should continue to adapt. I’m confused how they relate.

---

> > > ### Author Response · Authors · 2020-11-24
> > > **Reply II to Review by AnonReviewer1**
> > >
> > > Thank you for the comments and questions. Here are our answers:
> > >
> > > **“[...] a lifetime is in fact defined to include multiple episodes.”**
> > >
> > > Sorry, if our description is not as clear as it should be. A lifetime is equivalent to the length of one full episode. After a transition to an object location, the agent is ‘teleported’ back to the initial state in the bottom left corner (see section 4, p. 7). We added the following sentence to the manuscript, in the hope that the setup clearer now:
> > >
> > > *Within one episode,  the agent can hence first perform one or several exploration runs, in which it identifies the object location, and then do a series of exploitation runs, in which it takes the shortest path to that location.*
> > >
> > > Practically, this means that both the hidden state and the timestamp input t are not reset between ‘teleportations’. Instead, the agent goes through one single long episode similar to the setup in Micconi et al. (2020, section 4.2). A multi-episode technique such as VariBAD (Zintgraf et al., 2019), on the other hand, would terminate an episode after a set of fixed timesteps (say 15) and restart a new episode. This termination would effectively reset the exploration and potentially make the problem easier for the agent. In our case the agent has to “manage” their entire lifetime.
> > >
> > >
> > > **“[...] It seems completely expected that if the value of “learning” or “not learning” is the same, then the agent is free to pick one strategy or the other. While the inductive bias matters at the boundary, it doesn’t matter farther from the boundary, right?”**
> > >
> > > Yes, on the edge, both behaviors yield the same reward, but the parameter set that we checked is in fact in a regime where learning would be optimal. We now clarify this in the test:
> > > To test this, we trained $N=1000$ networks with different initial conditions, for task parameters close to the edge, but in the regime where the theoretically optimal strategy would be to learn.
> > > For this setting, the values for the two strategies are not the same, but only close to each other (see figure 2, top row, middle column). Importantly, the policy that achieves the value of “learning” is very different from the “not learning” policy. Our results show that the agent can get stuck in the local optimum of “not learning”, when in fact “learning” is optimal.
> > >
> > >
> > > **“[...] I think the idea is that in the approximate case, we might encounter local maximums that are not actually optimal (?) If this is the case, I think this problem needs to be demonstrated empirically to ensure it exists.”**
> > >
> > > We agree, and that is the purpose of the new Figure 2. In the bottom row of this figure, we therefore train 1000 independent networks with different initializations to validate that the observed bimodality is not a statistical fluke. We find that for settings that are far from the edge (left and right column in Fig. 2), the optimal behavior is meta-learned by all networks. For the case in which learning is theoretically optimal - but we are close to the non-learning regime - we find that a bimodality. Approximately 37% of all seeds end up in the local optimum of the deterministic policy. The remaining seeds learn the global optimum of a learning strategy.
> > >
> > >
> > > **“[...] The first part of this paragraph is I think referencing the fact that different solutions are possible at the learning/ non-learning boundary. The second half seems to be referencing a setting where an agent is trained for short-horizon and thus adapts quickly, but then is tested in a longer-horizon setting where it should continue to adapt. I’m confused how they relate.”**
> > >
> > > Please excuse the confusion. We have changed the paragraph in the manuscript for further clarification:
> > >
> > > *“It is beneficial to ensure rapid adaptation on the real robot, e.g., to avoid physical damage. To achieve this, the meta-learner should be optimized for a short horizon. This, however, introduces a bias towards not learning, or in more complex settings, for heuristic solutions that explore less than is required to discover the optimal policy. Hence, for such problems, the curation of the meta-learning task ensemble may have to additionally take into account potential nonlinear and long-lasting trade-offs between final performance and speed of adaptation.”*
> > >
> > > This statement is related to the risk of lacking generalization across horizons, but we argue that it can be exacerbated by the observed nonlinear dependence of the policy on horizon, because the meta-learner could potentially learn a very suboptimal policy for a short horizon.

---

### Official Review · AnonReviewer3 · 2020-10-29
**Review of "Learning not to Learn" for ICLR 2021**

**Rating:** 6
**Confidence:** 4

**Review:**

**Summary.** The authors investigate the question of when the optimal behavior for an agent is to learn from experience versus when the optimal behavior is to apply the same (memorized) policy in every scenario. They begin by introducing a simple bandits environment wherein they derive the optimal policy and identify regimes in which it involves memorization vs. learning. Then they train an RL^2 agent and verify that it behaves as expected in these regimes. Next, they expand their approach to a slightly more complicated gridworld environment which does not have an analytic solution to the question. The agent behaves as expected in the gridworld environment.

**Strong points.** This paper tackles a novel question which is fundamental to the field of metalearning. By carefully analyzing the two regimes of learning and memorization in the context of metalearning, this paper will increase awareness about the fact that the two regimes exist. The paper is clearly written and does an excellent job of putting experiments in the context of past ML research. The experimental setup is simple but goes straight to the heart of the issue. Figures and text do a good job of analyzing results and communicating them to the reader. Overall, this paper was an interesting read.

**Weak points.** The idea that “sometimes memorization is best and other times learning is best” does border on the obvious. Indeed, as soon as the authors derive their analytical solution, it becomes clear that we can expect the RL^2 agent to learn the same behavior. For me, there were no surprises in the experimental sections. To the authors: was there anything that was surprising or not obvious to you? What additional information can the experiments tell us, apart from confirming theoretical predictions?

Having said that, I also believe that very simple, well-executed research ideas sometimes make the best papers. This paper appears to be one of those cases. And even though the ideas are simple, they are significant and they are not a major part of the dialogue in the meta-learning community yet. So even if the ideas seem obvious, I think there is value in communicating them well.

I have one concern about the bandit task setup: the authors adjust $\sigma_l$, the width of the Gaussian from which they are sampling the reward, as a proxy for aleatoric uncertainty and hence task complexity. In doing so, they essentially equate “stochasticity of the environment” with “task complexity.” And yet, there are many other ways in which a task can be complex. Sometimes, all the information needed to perform a task is present and yet the task is difficult to solve because one needs to interpret/integrate the information in a particular way. This is why, for example, puzzles are considered difficult tasks. It is also why simulating the 3-body problem is a complex task. To the authors: can you clarify what you mean by “task complexity”?

In the closing paragraph of the paper, the authors claim that their approach “allows us to study the emergence of inductive biases in biological systems” but this claim is not supported by the rest of the paper, which makes almost no connections to biological systems. There are certainly ways in which these results are relevant to learning in biological systems, but the authors did not explore them in this paper, and so this claim is not well supported. In the same paragraph, they bring in contrasting notions of Darwinian and Lamarkian inheritance. Since they do this in one sentence -- the last sentence -- it is hard to understand what their claim is. And it was not clear that this was one of the main takeaways of the paper, as these concepts do not appear anywhere else in the paper. If the authors want to draw these conclusions, then they should add additional discussion on these topics. Otherwise, they risk misleading readers.

One additional minor suggestion would be to invert the color scale of Figure 6, as “white -> red” signifies values of increasing size in all preceding plots, but in Figure 6 it currently signifies values of decreasing size.

Minor grammatical suggestions
-- “the question which aspects of behavior“ -> “the question of which aspects of behavior“
-- When typing quotes in LaTex, use `` and ‘’ instead of “” so as to make them open & close correctly
-- “interplay of the agent’s lifetime,” -> “interplay between the agent’s lifetime,”
-- “We numerically show” -> “We show numerically”
-- “as well as explicit models of memory” -> ”and explicit models of memory” (same issue occurs later)
-- “the agents does not have” -> “the agent does not have”

**Recommendation.** 6 : Marginally above acceptance threshold

**Reasoning.** This paper is well written and the experimental setup is simple, well-executed, and produces results that are relevant to the main question of the paper. The main question of the paper -- when does it make more sense to learn vs. memorize a behavior -- is significant to ICLR and to the field of machine learning. There are a number of relatively minor weaknesses (as described above) but this is overall a nice paper and would be a good contribution to ICLR 2021.

---

> ### Author Response · Authors · 2020-11-18
> **Reply to Review by AnonReviewer3**
>
> Thanks, we very much appreciate your thoughtful considerations as well as detailed corrections. We want to highlight some new analyses and comments to your points of concern:
>
>
> **“[...] For me, there were no surprises in the experimental sections. To the authors: was there anything that was surprising or not obvious to you?”**
>
> Honestly, we were initially surprised by the fact that the transition to the non-learning parameter regime is an abrupt one, and we believe that this has consequences for meta-learning on a more general level. The underlying reason for the sharp transition is a bimodality in the reward landscape. The transition occurs where the global optimum switches from one of those local maxima to another, as for most phase transitions. As a result, the meta-learned strategy has a highly nonlinear and discontinuous parameter dependence and for gradient-based approaches, local minima are obviously an issue.
>
> To highlight this point, we now provide a further in-depth analysis of the transition regime between learning and non-learning. We show that in this regime, different initializations lead to fundamentally different solutions. We have added a completely new figure 2 including a discussion (please see reply to comment 2 of R2), as well as a discussion paragraph:
>
> *"[...] A key take-home from our results is the highly nonlinear and potentially discontinuous dependence of the meta-learned strategy on the parameters of the task ensemble. For certain parameter ranges, the reward landscape of the meta-learning problem features several local maxima that correspond to different learning strategies. The relative propensity of these strategies to emerge over the course of meta-learning depends on the task parameters and on the initialization of the agent. Generally, this supports the notion that there is not a single inductive bias for a given task distribution. Rather, there could be a whole spectrum of inductive biases that are appropriate for different amounts of training data. Even for the same task setting, different training runs can result in qualitatively different solutions, providing a note of caution for interpretations drawn by pooling over ensembles of trained networks." (p. 8)*
>
>
> **"[...] There are many other ways in which a task can be complex. Sometimes, all the information needed to perform a task is present and yet the task is difficult to solve because one needs to interpret/integrate the information in a particular way."**
>
> We agree, and tried to be a bit more precise in our definition of task complexity. An exploration of different types of task complexity is beyond the present paper:
>
> *"Task complexity: How long does it take to learn the optimal strategy for the task at hand? Note that this could be different from the time it takes to execute the optimal strategy." (p. 2)*
>
> We have also changed the description of the bandit task to better explain our choices and clean up the admittedly confusing terminology in the earlier version of manuscript. Here the relevant excerpt for task complexity:
>
> *"While the mean µ remains constant for the lifetime T of the agent, the reward obtained in a given trial is stochastic and is sampled from a second Gaussian,  r ~ N(µ, σ_l). [...] The standard deviation σ_l hence controls how quickly the agent can learn the optimal policy. We therefore use it as a proxy for task complexity.” (p. 3)*
>
>
> **“[...] There are certainly ways in which these results are relevant to learning in biological systems, but the authors did not explore them in this paper, and so this claim is not well supported.”**
>
> We also agree that the claim of studying inductive biases in real biological systems is too vague and have significantly adapted the manuscript. Nonetheless, we believe that future in-silico meta-learning approaches can provide testable predictions for experimental evolution. This includes the interesting line of work on experimental evolution such as simulating up to 50 generations of drosophila evolution (e.g. Dunlap and Stephens, 2016; Marcus et al., 2018). We added a discussion paragraph and scaled down our claims:
>
> *"[...] From this perspective of animal behavior, meta-learning with a finite time horizon could provide an inroad into understanding the benefits and interactions of instinctive and adaptive behaviors. Potential applications could be the meta-learning of motor skills in biologically inspired agents (Merel et al., 2019) or instinctive avoidance reactions to colours or movements. The degree of biological realism that can be reached will be limited by computational resources, but qualitative insights could be gained, e.g., for simple instinctive behaviors." (p. 9)*
>
> Finally, we thank the reviewer for the detailed corrections. We apologize for the grammatical faux pas and have corrected these as well as adapted the color scheme of figure 6 in the current version of the submission manuscript.

---

> > ### Comment · AnonReviewer3 · 2020-11-23
> > **Updates to paper represent an improvement; continued concerns about significance**
> >
> > I read the latest draft of the paper as well as the authors' responses above. They have addressed my three most significant concerns by adding better discussion to the paper and responding to this review directly in a clear manner. But even given those updates, I stand by my original position: _The idea that “sometimes memorization is best and other times learning is best” does border on the obvious. Indeed, as soon as the authors derive their analytical solution, it becomes clear that we can expect the RL^2 agent to learn the same behavior. For me, there were no surprises in the experimental sections._ This position is echoed by the other reviewers:
> > * R1: "...results are intuitive and unsurprising, they nicely emphasize the importance of environment and task"
> > * R3: "My main criticism of the work is that the results seem fairly obvious. In the finite horizon case, the meta-learner will necessarily learn a strategy optimal for the given horizon because that’s exactly what it’s optimized for."
> > * R4: Comments on how previous works can be leveraged to make the same conclusions
> >
> > To summarize,
> > **Pros.**
> > * Work is thorough, clear, and well-written. Experiments are solid. Authors have responded well to our comments and concerns.
> > * Worth drawing attention to the "learning not to learn" regime in metalearning
> > * R4: "This paper is generally well-executed and clear."
> >
> > **Cons.**
> > * Results are unsurprising
> > * Unclear what the significance of the work is to the community
> > * R4: "Overall, it is therefore unclear to me what significant takeaways the meta-RL community can gain from this work."
> >
> > As I wrote in my original review, I believe that the merits of the paper -- experiments are well-done, writing is clear, and it raises awareness about an important aspect of metalearning -- make it worth accepting to ICLR. I maintain my score of 6.

---

### Official Review · AnonReviewer2 · 2020-10-31
**Official Blind Review AnonReviewer2**

**Rating:** 7
**Confidence:** 4

**Review:**

This paper provides an analysis of RNN-based meta learning approaches. In particular, it investigates the strategies learned via meta-learning, contrasting strategies involving task-dependent learning vs heuristic or hard-coded solutions. Empirical evidence in two sets of experiments, on a 2-armed bandit toy task and a grid-world navigation task, show that hard-coded strategies can be a function of training task distribution and task complexity as well as task horizon.

While the experiments are simplistic, they provide a clear and thorough comparison of different agent behaviours across different training regimes. I enjoyed reading the paper and although the results are intuitive and unsurprising, they nicely emphasize the importance of environment and task design choices in strategies learned via memory-based meta-learning.

Comments/questions:
1. I would encourage the authors to use “memory-based/RNN-based meta-learning” instead of “meta-learning” to avoid confusion as these results might not apply more widely across different meta-learning approaches (e.g. gradient based).
2. While the pattern of behaviour looks qualitatively similar across analytical and empirical results reported in Figure 1, I wonder if it would be possible to quantitatively assess where they differ.
3. It would be nice to see error bars for the 5 independent training runs in Figure 5.
4. In the Appendix A.2.3 it is mentioned that the discount factor is annealed from 0.8 to 1. within the first 800k episodes. Could you please expand if this was crucial to achieve your results or just an experimental choice?

Overall, I think this is an interesting contribution of perhaps limited scope but still valuable and could encourage interesting future research directions in memory-based meta-learning.

---

> ### Author Response · Authors · 2020-11-18
> **Reply to Review by AnonReviewer2**
>
> Thank you very much. We have modified the manuscript and hope your questions and suggestions are now addressed. Please see below for detailed replies:
>
> 1. **"[...] I would encourage the authors to use “memory-based/RNN-based meta-learning” instead of “meta-learning” to avoid confusion as these results might not apply more widely across different meta-learning approaches (e.g. gradient based)."**
> Yes, we fully agree. The previously used meta-learning terminology may be confusing for the reader and we have replaced the wording to be more precise in the current version of the submission. (See revised manuscript.)
>
>
> 2. **"[...] While the pattern of behaviour looks qualitatively similar across analytical and empirical results reported in Figure 1, I wonder if it would be possible to quantitatively assess where they differ."**
> We believe that this is a very important question and have further investigated the differences between analytical and empirical solutions. Given that those occur primarily on the edge between the learning and the no-learning regimes, we did a deeper analysis of this parameter regime. We now show that in that regime, the reward landscape has local maxima into which the gradient-based optimization falls. How often which local maximum is reached depends on the distance of the parameters from the edge. The results can be found in a new figure 2. We also added a new paragraph discussing our findings:  *"[...] In the Bayesian model, the edge between the two regimes is located at parameter values where the learning strategy and the non-learning strategy perform equally well. Because these two strategies are very distinct, we wondered whether the reward landscape for the memory-based meta-learner has two local maxima corresponding to the two strategies (figure 2).  To test this, we trained $N=1000$ networks with different initial conditions, for task parameters close to the edge. We then evaluated for each network the number of explorative pulls of the stochastic arm, averaged across 100 episodes. The distribution of the number of explorative pulls across the 1000 networks shows i) a peak at zero exploration and ii) a broad tail of mean explorative pulls (figure 2), suggesting that there are indeed two classes of networks. One class never pulls the stochastic arm, i.e., those networks adopt a non-learning strategy. The other class learns. For task parameters further away from the edge, this bimodality disappears." (p. 5)*
>
>
> 3. **"[...] It would be nice to see error bars for the 5 independent training runs in Figure 5."**
> We added median and percentile bands to both figure 5 and 6. The results remain robust and significant. (See revised figures in the manuscript.)
>
>
> 4. **"[...] In the Appendix A.2.3 it is mentioned that the discount factor is annealed from 0.8 to 1. within the first 800k episodes. Could you please expand if this was crucial to achieve your results or just an experimental choice?"**
> Indeed, the discount factor schedule is an important hyperparameter. While our theoretical result relies on a finite horizon and discount factor 1, this setting is initially very challenging for the meta-learner, the main reason being the long time-scale of backpropagation through time when training the RNN. Instead, we chose to gradually extend the window of integration by increasing the discount factor. Thereby, we essentially treat the discount as a curriculum parameter (Prokhorov and Wunsch, 1997).

---

> > ### Comment · AnonReviewer2 · 2020-11-23
> > **Improved paper**
> >
> > I thank the authors for responding to my concerns and improving the clarity of the paper.
> >
> > After reading the other reviews and taking into account the additional details provided by the authors, I kept my score but increased my confidence. I agree with the other reviewers that the experiments are unsurprising and the major finding of the paper resembles conclusions made in prior works showing that meta-RL can effectively learn the Bayes-adaptive optimal policy. However, I still believe that the work would be interesting to the community and can inspire more thorough investigation of the effect of training task distribution on the strategies learned via meta-learning and I share the authors' concern that this issue has been overlooked in most prior work. The experiments are clear and the authors have done a good job of addressing some of the concerns raised by the other reviewers and further improved the paper.
> >
> > Overall, I would support publishing this paper, but I understand if the consensus goes the other way.

---

### Author Response · Authors · 2020-11-18
**General comments to all reviewers**

We thank the reviewers for the their constructive feedback, which - we think - has substantially helped us to improve the paper. We have revised the manuscript accordingly, and hope that their concerns have been addressed. A red-lined version of the revised manuscript has been uploaded.

---

### Decision · Program_Chairs · 2021-01-07
**Final Decision**

**Decision:**

Reject

**Comment:**

There was fairly detailed discussion among three of the four reviewers. The fundamental concern of the reviewers is regarding the contribution of the paper. During the rebuttal, the authors clarified the following:

> while the effects of varying uncertainty / horizon lengths is well-understood for Bayes-optimal policies, it is not understood for existing meta-RL approaches, which is the topic of this paper

That is, the contribution of the paper is to understand the effects of varying uncertainty/horizon lengths for meta-RL approaches. However, it is known in prior work that meta-RL algorithms such as RL^2 can implement Bayes-optimal policies in principle. As a result, it's not clear whether this contribution is significant relative to prior knowledge, and this paper does not seem to bring any new insights.

An alternative framing of the paper would be to consider the question of how meta-RL solutions compare to Bayes-adaptive optimal policies. While this framing would be interesting and novel, the current version of the paper does not sufficiently answer this question, since the only experiments include RL^2 (and such a study would require experimenting with more sophisticated meta-RL algorithms beyond RL^2).

As such, this paper isn't suitable for publication at ICLR in its current form.